# Deep neural networks integrating genomics and histopathological images for predicting stages and survival time-to-event in colon cancer

Olalekan Ogundipe[1], Zeyneb Kurt [2]*, Wai Lok Woo[1]

1 Department of Computer and Information Sciences, University of Northumbria, Newcastle Upon Tyne, United Kingdom, 2 Information School, University of Sheffield, Sheffield, United Kingdom

* z.kurt@sheffield.ac.uk

## Abstract

### Motivation

There exists an unexplained diverse variation within the predefined colon cancer stages using only features from either genomics or histopathological whole slide images as prognostic factors. Unraveling this variation will bring about improved staging and treatment outcomes. Hence, motivated by the advancement of Deep Neural Network (DNN) libraries and complementary factors within some genomics datasets, we aggregate atypia patterns in histopathological images with diverse carcinogenic expression from mRNA, miRNA and DNA methylation as an integrative input source into a deep neural network for colon cancer stages classification, and samples stratification into low or high-risk survival groups.

### Results

The genomics-only and integrated input features return Area Under Curve–Receiver Operating Characteristic curve (AUC-ROC) of 0.97 compared with AUC-ROC of 0.78 obtained when only image features are used for the stage's classification. A further analysis of prediction accuracy using the confusion matrix shows that the integrated features have a weakly improved accuracy of 0.08% more than the accuracy obtained with genomics features. Also, the extracted features were used to split the patients into low or high-risk survival groups. Among the 2,700 fused features, 1,836 (68%) features showed statistically significant survival probability differences in aggregating samples into either low or high between the two risk survival groups.

**Availability and Implementation:** https://github.com/Ogundipe-L/EDCNN

## 1. Introduction

There are over 67 cancer primary sites in the human body [1], among which are the brain, breast, colon and several other cancer types that have been identified and detected in various

---

**Data Availability Statement:** All datasets are fully available on publicly available secondary data repositories without restriction. Functional genomics datasets were downloaded from The

Cancer Genome Atlas (TCGA-COAD cohort, available on https://portal.gdc.cancer.gov/) and the histopathological images of the same TCGA cohort were downloaded from https://zenodo.org/records/2530835#.YzbcnuzMLpC.

**Funding:** The author(s) received no specific funding for this work.

**Competing interests:** NO authors have competing interests

parts of the body. The lung has the highest cancer cases and the colon ranks fourth. According to the World Health Organization 2020 report, cancer is the leading cause of death accounting for over ten million deaths (https://www.who.int/news-room/fact-sheets/detail/cancer)) with colon cancer accounting for 916 000 deaths. Available research tools and datasets provided by the cancer research community for analyzing causes and treatment of cancer types are genomics and histopathological whole slide images datasets among others, each consisting high-dimensional features embedding the complex pathological pattern of each cancer stages [2]. The cancer staging system determines the amount and spread of cancer in a patient's body and the most common practice is the use of the TNM (Tumor-Node-Metastasis) system (https://www.cancer.gov/about-cancer/diagnosis-staging/staging). T describes the size of the tumor and any spread of cancer into nearby tissue; N describes the spread of cancer to nearby lymph nodes; and M describes metastasis (spread of cancer to other parts of the body). This system was created and is updated by the American Joint Committee on Cancer (AJCC) and the International Union Against Cancer (UICC). The TNM staging system is used to describe most types of cancer. Higher performance measure and improved accuracy has been shown using a deep neural network to extract patterns from multimodal data sets for cancer staging compared with using the TNM signature often considered to be a subjective system [3–5]. The available deep neural networks still require the development of an efficient and effective model for accurate and improved classification and effective treatment management.

Colon cancer is a malignant tumor of the large intestine or of the rectum that affects both males and females irrespective of age group [6–9]. This cancer could occur by genetic mutations or through heredity in an individual. Hundreds of millions of cases have been diagnosed every year and tens of thousands of deaths reported globally every year [10]. In terms of mortality rate, colon cancer is ranked second [11, 12]. Cancer diagnosis can be classified into four stages ranging from I to IV where stage I implies that the cancer is in early stages and IV indicates that the cancer is in advanced stages (https://www.cancercenter.com/cancer-types/colorectal-cancer/stages).

One of the most challenging areas in cancer research is staging, which is an indicator of the patient's most likely outcome, life expectance and chances of cure [13–16]. Researchers studying colon cancer are working on various ways to unravel the means of prevention, treatment and timely detection of the disease to reduce the mortality and incidence rate as well as improve the quality of life for people infected with colorectal cancer [17, 18]. Deep learning computational framework has been deployed in the study and analysis of biospecimen data ranging from features encoding and extraction to integration and transformation of heterogeneous biological data [19–21]. Most of the previous studies only concentrated on unimodal data feature studies, and some that are based on multimodal data are either purely clinical trials [22–24], or use linear models and eigengenes to extract features as the baseline framework [25], or are earlier deep learning frameworks [26]. However, our study relies on bimodal fusion of features from genomics (DNA methylation, mRNA, and miRNA) and slide images cumulating into quadruple datasets for tumor stage examination using a deep learning framework. The analysis and result of our proposed model show a slight improvement in cancer stage prediction and 68% of the integrated features cluster the samples into high-low survival risk groups.

In recent research involving cancer diagnostics and prognostic outcomes, multimodal datasets have been adopted to improve prediction and prognostic accuracies. For example, Vale-Silva et al [27] extracts and integrates features from four different modalities for deep learning survival predictions. Complementary features are extracted from tissue biopsy imaging, copy number variation, gene, miRNA expression and DNA Methylation data. Elsewhere, Heo et al [28] established the use of multimodal imaging to achieve higher precision to determine the

**Table 1. Sample and feature size of genomics dataset before and after preprocessing.**

| Biological data/assayPlatform | Before Preprocessing | | After Preprocessing | |
|---|---|---|---|---|
| | No. of Samples | No. of Features | No. of Overlapping Samples | No. of Features |
| Clinical/ BiospecimenClinicalData | 448 | 80 | 255 | 80 |
| mRNA/ (gene.normalized_RNAseq | 328 | 20,502 | 255 | 16,377 |
| miRNA /(mir_Hiseq.hg19.mirbase20) | 261 | 1,870 | 255 | 420 |
| DNA methylation /(methylation_450) | 353 | 20,759 | 255 | 20,129 |

restaging of rectal cancer after chemoradiotherapy (CRT). Also, Olatunji et al [29]shows that a multimodal dataset is more predictive of distance metastasis (DM) than when an unimodal dataset is used. This research study focuses on improvement in the accuracy prediction of colon cancer into any of the four (I-IV) stages. The contributions of this research study are:

Design of a novel algorithm for data preprocessing conditions on identifying tumor quality within histopathological images and inclusion of genes within the genomics dataset that satisfy a certain set condition/requirement.

We are identifying and fusing complementary features from genomics and histopathological images to improve accuracy and performance in cancer stage prediction and survival risk stratification.

Training of DNN model with effective data rate and avoiding information redundancy through effective and robust data preprocessing

## 2. Materials and methodology

All datasets are fully available on publicly available secondary data repositories without restriction; no new data has been generated while undertaking this study. The data used in this study was downloaded from TCGA-COAD (https://portal.gdc.cancer.gov/), the Cancer Genome Atlas (TCGA) colon cancer cohort with TCGA-Assembler function 2. Five different sets of data relating to colon adenocarcinoma (COAD) colon cancer type were downloaded from their respective source (assayPlatform) shown in Table 1 are (Clinical, DNA methylation, miRNA, mRNA, Hematoxylin and Eosin (H&E) stained histopathological images). Based on our research plan, the first set of data download is the genomics which resulted in 448 samples of clinical records with 80 features, 328 samples of mRNA expression with 20,502 features, 261 samples of miRNA expression with 1,870 features, and 353 samples of DNA methylation with 20,759 features. After applying the comprehensive preprocess procedures as explained in section (2.2) on the genomics datasets 255 samples are estimated to have data present in (DNA methylation, miRNA, mRNA). We proceeded (https://portal.gdc.cancer.gov/) to get the Hematoxylin and Eosin (H&E) stained histopathological images of colon cancer for the 255 samples that have data in the genomics but got only 177 equivalent samples as in the genomics with images datasets. See Tables 1 and 2 for details in sample sizes.

**Table 2. Sample size distribution across whole slide images and genomics data.**

| Whole slide images | | | Genomics sample size | | |
|---|---|---|---|---|---|
| Stages | Sample size | Number of Tiles | mRNA | miRNA | DNA Methylation |
| I | 30 | 27,358 | 30 | 30 | 30 |
| II | 67 | 47,397 | 67 | 67 | 67 |
| III | 54 | 21,492 | 54 | 54 | 54 |
| IV | 26 | 15,914 | 26 | 26 | 26 |
| | 177 | 112,161 | 177 | 177 | 177 |

## 2.1 Genomics datasets preprocessing

We develop a preprocess model parameterized by the datasets with the output satisfying the following criteria: (i) First, a biological feature (in any of methylation or mRNA or miRNA data) is removed, if more than 20% of the patients have a 0 value for it. (ii) A sample is removed if more than 20% of its features are missing. (iii) Then a substitution function in python is used to fill out the missing values with zero. (iv) Only common samples (individuals) existing in all the data sets (mRNA, miRNA, methylation) are kept. (v) The data sets (mRNA, miRNA, methylation) are individually normalized with z-score. (vi) The z-score normalized data are combined and re-normalized with unit scale (L2-norm) transform.

## 2.2 H&E histopathological image preprocessing

The stained hematoxylin and eosin whole slide images downloaded from The Genome Atlas (TCGA-COAD) of colon cancer are large in size with an average of 2GB and high resolution. Each of the 177 whole slide image samples used in the study was divided into several tiles of size 224 x 224 pixels with openslide-python packages, we built a sub-python function called MX to filter out patches with less than 30% cellular tumor content. The MX function selects all patches with cellular content equal to or greater than 30% through the thresholding method by estimating the patches' background noise and the foreground tumour content. The thresholding procedure entails converting the patch images in RGB to grayscale color and segmented into background and cell component regions. Patches with foreground content equal to or greater than 70% are selected for training the prediction model. After preprocessing a total of 112,161 image tiles satisfy the conditions and requirements set out for the research study. Salient features are extracted from the resulting 112,161 image samples and integrated with equivalent genomics features extracted from combinations of mRNA, miRNA and DNA methylation.

## 2.3 Features extraction from (mRNA, miRNA, DNA methylation)

We concatenate 16,377 features from mRNA, with the 420 features and 20,129 from miRNA and DNA methylation, respectively to obtain a multimodal dataset with 36,926 features. We input the 36,926 merged features from miRNA, mRNA and DNA Methylation into an autoencoder (AE) neural network designed specifically to encode its input. The AE leverages a scalable hyperparameter optimization framework that searches the AE space for best hyperparameter values for a sequence of bottleneck encoding features that will ensure optimal performance of the AE network for the classification and stratification purposes as proposed in [30]. After several trials, 652 extracted features give an optimum cancer stage prediction accuracy.

## 2.4 Features extraction from Hematoxylin and Eosin (H&E) stained histopathological images

Features extraction from the image patches is carried out using transfer learning and fine-tuning processes with ResNet50 deep neural network based on the standard transfer learning workflow. Following the standard workflow: First, pretrained weights from ResNet50 are loaded into a defined base model, then, the layers in the base model freeze. We then defined a new predictor on the based model, we defined a new model on base model with new predictor and trained the new model on some of the histopathology image tiles across the four cancer stages. Next, we used the new model with weights adapted to histopathological images for the extraction of 2,048 salient features from each of the 112,161 image tiles. The ResNet50 preprocessing input aid to zero-center each color channel for each image tile without scaling during the extraction process.

## 2.5 Integration of features extracted from histopathological images and genomics

Methods for combining vector representation from histopathological and genomics include an element-wise product or sum, Multimodal Compact Bilinear pooling (MCB) [31] and concatenation. Bilinear pooling computes the outer product between two vectors, which allows, in contrast to the element-wise product, a multiplicative interaction between all elements of both vectors. On the other hand, the concatenation method linked the two representations in series. Individually, each type of data adopted carries a different mutated and complex topological outlay of cancerous tumor, each with limited amount of biology carcinogenic factors. We conjectured combining them would provide more complementary and comprehensive prediction features. We aggregate the encoded features in each data by concatenating extracted 652 features from the genomics and 2048 features from histopathological images to create new multimodal data with 2700 features for each sample as represented in Fig 1.

The feature extraction process from image tiles and the genomics datasets concluded with the dimensions of 112,161(samples)x2048(dimensions) and 177(samples)x652(dimensions), respectively. We used a many-to-one technique to concatenate vectors from the images and genomics datasets, requiring that concatenated features from both sides (image and genomics) have the same patient identification code. The concatenation procedure generates a fused dataset of size 112,161(samples)x2700(dimensions).

To guarantee that an equal number of rows from each of the three modality options (i.e. (i) images only, (ii) genomics only, (iii) fused image and genomics) are utilized for training the prediction models, the complete fused dataset is labeled as the integrated dataset with dimensions 112,161x2700.

The datasets with 112,161 rows in each modality were divided into 89,729 (80%) for the training and 22,432 (20%) for model testing. At the patient's level, this represents 142 (80%) samples for training and 35 (20%) samples for testing.

The analysis and result of the study were based on the testing portion of the datasets, equivalent to 22,432 (20%) in each modality.

## 2.6 Deep Neural Network (DNN) implementation

To implement the first two steps of the proposed DNN which are features extraction processes, we chose two network models which are sparse autoencoder (AE) [30] and ResNet50 [32]. The sparse autoencoder is adopted for joint encoding and merged features from mRNA, miRNA and DNA methylation using combinations of non-linear functions. The resulting encoded genomics features are embedded with robust phenotypic nomenclature more definitive for the cancer staging and survival risk stratification. The second model, ResNet50—a 50-layer residual deep neural network was also used for feature extraction from the histopathological images.

ResNet50 is the 50 layers of ResNet framework proposed by [32]. ResNet is a residual learning mechanism embedded within the deep Convolutional Neural Network (CNN) to make it more effective. The residual training mechanism is designed to handle dual problems of vanishing gradients and degradation of training accuracy. The three functional layers within neural network activating the extractions of salient features from image tiles, are the convolution, pooling and average-pooling layers. Features from images are extracted as proposed in [33].

## 2.7 Training, evaluation and testing setup

To prevent overfitting and ensure equal proportions of samples per stages are selected in each batch of data used in training the classifier, the datasets is divided into training and validation

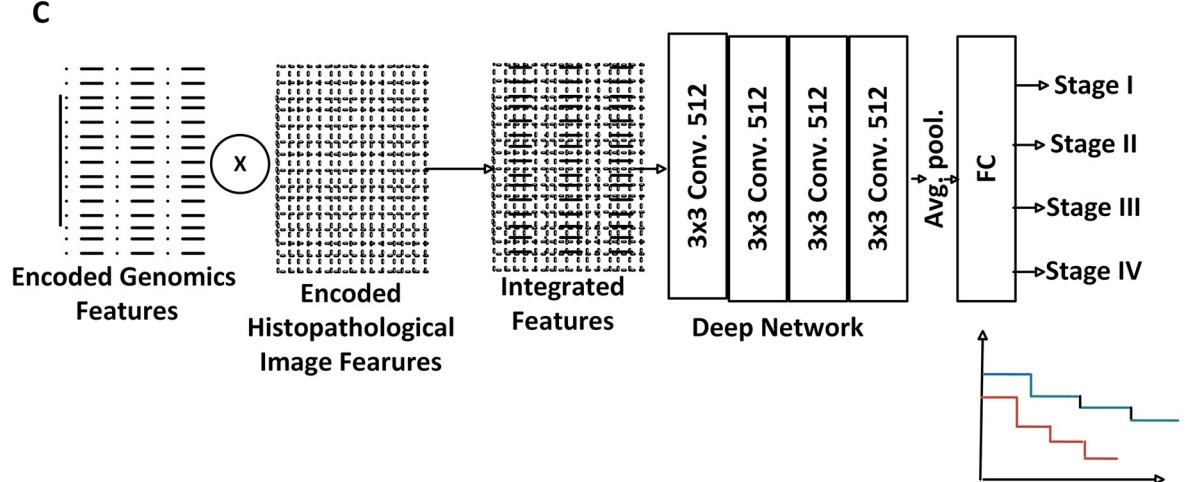

**Fig 1.** The proposed DNN model with three major steps (A) extraction of salient features from concatenated biological genes of mRNA, miRNA and DNA- Methylation (B)feature extraction from histopathological images (C) CNN stages prediction with fused features.

**Table 3. Number of samples (i.e. image tiles) per cancer stage for training the predictive model.**

| Sample Stages | No. of samples available for predictive model | % per stage |
|---|---|---|
| Stage1 | 27,358 | 24.39 |
| Stage2 | 47,397 | 42.26 |
| Stage3 | 21,492 | 19.16 |
| Stage4 | 15,914 | 14.19 |

set 80% and testing set 20%. The 80% portion for the training and validation is further divided into training set 80% and validation set 20% using a stratified 5-fold method. According to distribution of the datasets as shown in the Table 3, the dataset is imbalanced across all the stages with samples in stage1 having 24.39%, stage 2 with 42.26%, stage 3 with 19.16% and stage 4 with 14.19%. We adopted 5-fold stratified sampling to preserve the stage frequency across each train and validation fold. During training Stratified 5-fold sampling and batching procedure are adopted to ensure randomness and preserve the stage frequency across each training and validation session.

The unimodal and multi-modal features are trained under the same experimental condition. The classifier was trained thrice, the first and the second training was done with encoded genomics and histopathological image data as input, while the third training used input from the integrated features of the encoded genomics and histopathological image.

## 2.8 Classification model and evaluation metrics

We hypothesized that synthesizing and aggregating extracted features from histopathological images and genomics Fig 1A and 1B as input data into CNN Fig 1C for colon cancer stages prediction and risk stratification may lead to an improvement and more accuracy in prediction and survival risk stratification. We test the validity of our hypothesis after reducing the dimension and consequently learn a lower-dimensional and compressed representation from the genomics and histopathological images.

The CNN classifier comprises of different layers including: the input layer, convolutional layers, pooling layers, and the output layer. The classifier is parameterized with categorical cross-entropy loss function, Adam optimizer (a replacement optimization algorithm for stochastic gradient descent for training deep learning models) and metric accuracy. The training and validation fitting is set to run for 100 epochs.

The experiment was designed and implemented on a server with ubuntu operating system fitted with 3 sets of NVIDIA GeForce GTX 1080 Ti GPU devices with a memory size of 10410 MB per GPU device. The entire code used during the implementation are python based on TensorFlow, Pandas, and scikit-learn libraries. Our proposed model is used to train each group of features. Two unimodal classifications, one based on image features and the other based on genomics features with the third classification based on fused features from images and genomics, and we measure the accuracy of DNN cancer stage prediction with Area Under the Curve Receiver Operating Characteristic (AUC ROC) metrics.

## 3. Results

### 3.1 Deep neural network for cancer stage prediction with unimodal and multimodal datasets

Following the earlier data preprocessing, ResNet50 transfer learning and fine-tuning feature extractions from histopathology images and features extractions from genomics datasets with stacked autoencoder. We did feature integration on extractions from histopathological images

**Table 4. AUC-ROC value for each cancer stages under each category of features.**

| Cancer stages | AUC based on features from images (% accuracy) | AUC based on features from genomics (% accuracy) | AUC based on features from WSI + genomics (% accuracy) |
|---|---|---|---|
| I | 0.79 | 0.97 | 0.97 |
| II | 0.76 | 0.97 | 0.97 |
| III | 0.76 | 0.97 | 0.97 |
| IV | 0.79 | 0.97 | 0.97 |

and genomics by concatenating of the extracted features from images and genomics from the same sample.

We train the cancer stage predictive model with the unimodal and integrated features followed by survival risk stratification based on the integrated features. For each group of data 80% of the samples are used for training and validation while the remaining 20% was set aside as testing data.

The training and validation section adopted 5-fold stratified methods on the 80% dataset meant for that purpose. Each fold runs for 100 epochs and accuracy measured as average over the 5-folds.

Performance and accuracy metrics are measured with AUROC Table 4 and confusion matrix Table 5 based on 20% test data never seen before by the learning predictive model. AUROC result (Fig 2A–2C) for stage prediction based on image features gives area under the curve (AUC) as 0.76 accuracy in stages 2 and 3 predictions and 0.79 AUC accuracy in predicting stages 1 and 4 cancer stages. Predictive accuracy with genomics features returns 0.97 AUC across all the four cancer stages. Likewise, predictive accuracy with integrated features from genomics and images returns 0.97 AUC across all the four cancer stages. Further analysis of the result computed with confusion matrix shows that there is 0.08% improved prediction accuracy using the integrated features from images and genomics datasets.

**Table 5. Confusion matrix analysis for unimodal and multimodal features estimated on test data which is 20% of the entire samples.**

| Images | Confusion Matrix | | | | | | Samples correctly predicted all stages | %Accuracy | %Improvement in accuracy |
|---|---|---|---|---|---|---|---|---|---|
| | True Stages | Stage1 | 2912 | 995 | 1134 | 430 | | | |
| | | Stage2 | 1121 | 5486 | 2157 | 716 | | | |
| | | Stage3 | 527 | 964 | 2436 | 369 | | | |
| | | Stage4 | 277 | 814 | 697 | 1395 | | | |
| | | | Stage1 | Stage2 | Stage3 | Stage4 | 12229 | 54.52 | |
| | | | Predicted Stages | | | | | | |
| Genomics | True Stages | Stage1 | 3825 | 1540 | 19 | 87 | | | |
| | | Stage2 | 0 | 9374 | 106 | 0 | | | |
| | | Stage3 | 0 | 1007 | 3291 | 0 | | | |
| | | Stage4 | 0 | 813 | 0 | 2370 | | | |
| | | | Stage1 | Stage2 | Stage3 | Stage4 | 18860 | 84.08 | 29.56 |
| | | | Predicted Stages | | | | | | |
| Integrated | True Stages | Stage1 | 3823 | 1619 | 26 | 3 | | | |
| | | Stage2 | 0 | 9395 | 80 | 5 | | | |
| | | Stage3 | 0 | 1007 | 3291 | 0 | | | |
| | | Stage4 | 0 | 813 | 0 | 2370 | | | |
| | | | Stage1 | Stage2 | Stage3 | Stage4 | 18879 | 84.16 | 0.08 |
| | | | Predicted Stages | | | | | | |

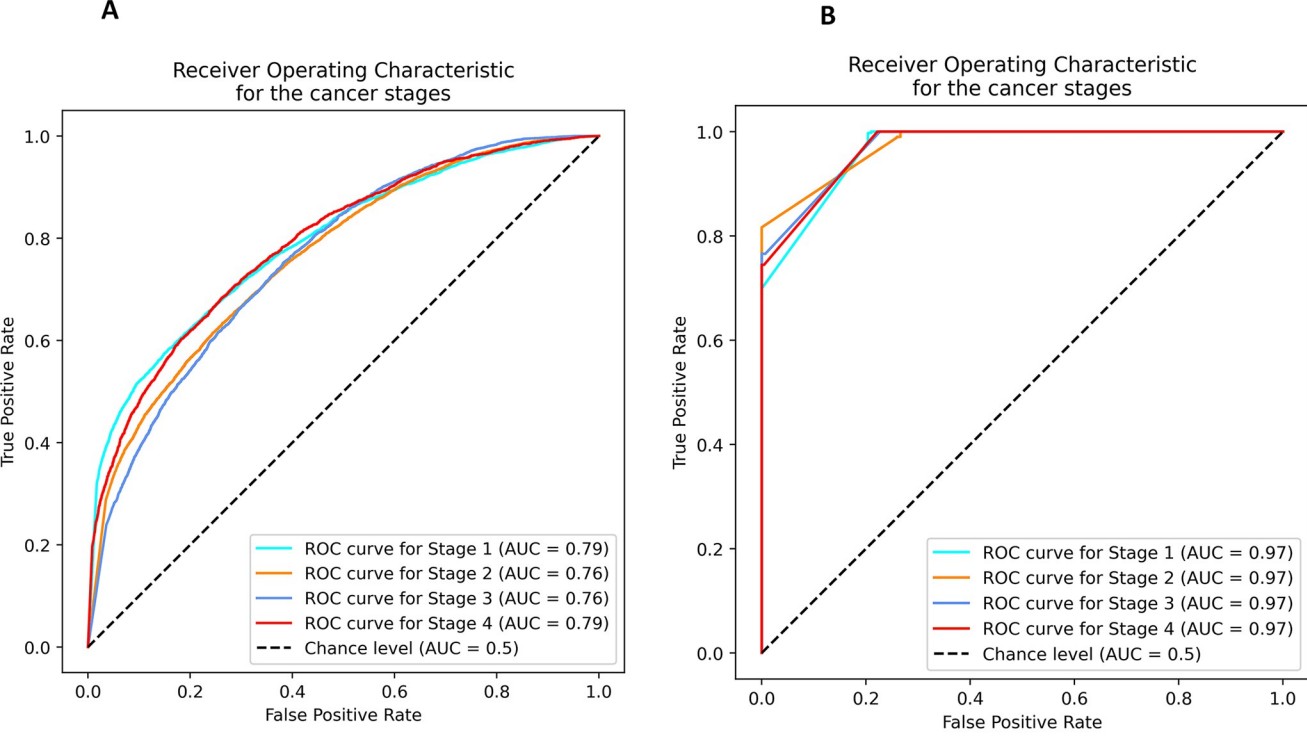

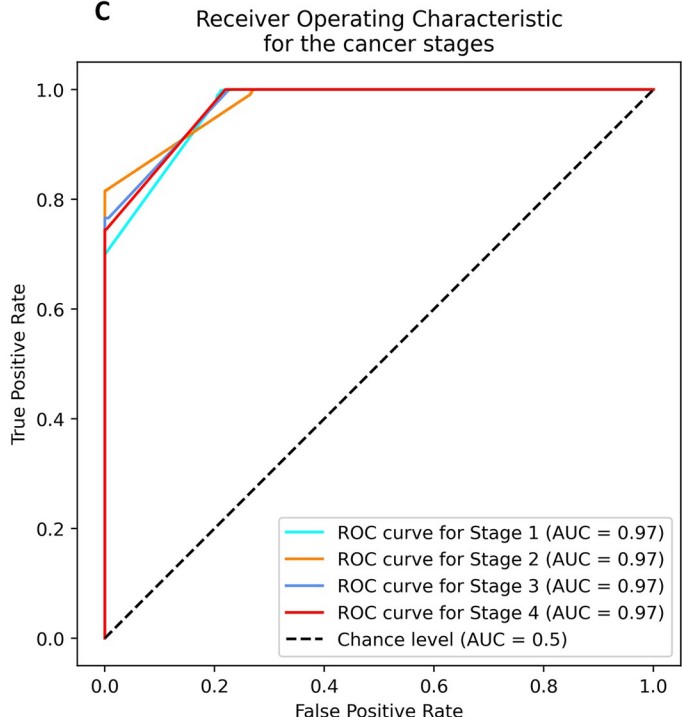

**Fig 2.** The predictive model AUC results using different input features (A) images only features (B) genomics only features and (C) Integrated features images and genomics.

## 3.2 Integrated latent space and survival analysis

We analyzed the proportion of the fused 2,700 extracted features that will group the patients into low and high-risk survival groups necessary for assessing the survival outcome. The survival analysis was built around three vectors representation. The first vector being the fused features and the other two vectors—vital status (indicating when the sample is alive or dead) and last contact days are from clinical record of patient with colon cancer disease. To achieve our set objectives, the median value of each extracted feature was estimated, and sample value under each feature classified as either belonging to low or high-risk survival group compared with its median value. The Kaplan-Meier estimator was used to estimate and visualize the difference between the low and high-risk survival sample groups under each feature individually and compare-survival algorithm of the scikit-survival library is used to compute the p-values related to statistical significance between the low and high-risk groups. The Kaplan-Meier method is an estimator for analyzing time-to-event data whose survival-time covariates are right censored, while the compare-survival function is the K-sample log-rank hypothesis test of identical survival functions. It compares the pooled hazard rate with each group-specific hazard rate. The alternative hypothesis is that the hazard ratio of at least one group differs from the others at some time. To control the probability of committing a type I error among the n = 2700 statistical test, we adopt Bonferroni Correction (BC) for adjusting the alpha level value which was originally set to 0.05. The BC is given as:

$$\alpha_{new} \leq \frac{\alpha_{original}}{n} \tag{1}$$

Where $\alpha_{newl}$ is the adjusted alpha level, $\alpha_{original}$ is the original alpha level and $n$ is total number of statistical tests perform. This translates into a situation that we reject the null hypothesis of each test within the multiple statistical tests only when the raw p-value is less than

$$\alpha_{new} \leq {\alpha_{original}}/{n} = {0.05}/{2700} = 1.8519e^{-5}. \tag{2}$$

Analysis indicates that 1,836 of the 2,700 extracted features are statistically significant.

## 3.3 Survival risk stratification with extracted features

The Kaplan-Meier estimator for some most significant extracted features showing the survival functions of patient in low and high-risk survival. Fig 3. gives the visual representation of the relationship between time and the probability of a patient in low or high-risk class surviving beyond a given time point. As shown in Fig 3, the wide gap between the two functions is an indicator that we can confidently argue that 68% of the extracted features conveniently group the samples into low or high-survival class. The result of p-values (showing the top five most significant features from genomics and images individually) by the log-rank statistical test support that majority of the extracted features can stratify the patients into low and high-risk survival groups is as shown in Table 6. These features with statistically significant p-value rejects the null hypothesis stating that there is no statistically significant difference between the high and low risk survival clusters. Also, the significance of the estimated p-values supports the evidence that the extracted features can stratify the samples into low and high-risk survival groups required and necessary for clinical prognosis. The 1836 stratification features consist of 1204 and 632 of image and genomics features respectively. We show the top 10 distinguishing features (five genomics, five image features) in Table 6 in terms of their capability to present the survival stratification.

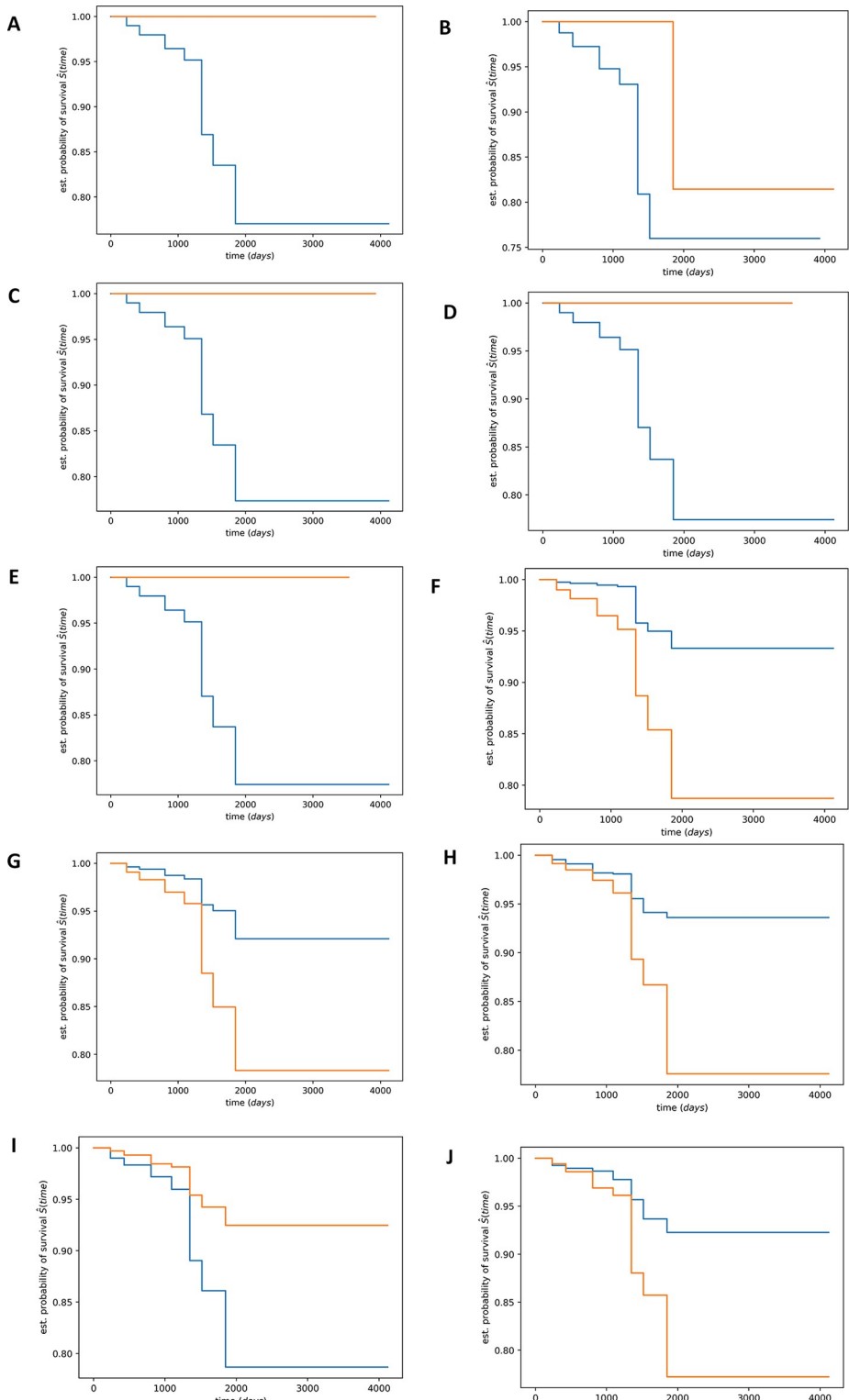

**Fig 3. Kaplan Meier samples stratification into low or high-risk survival group according to the most distinguishing 10 features among 1,836 features with statistically significant p-values Fig 3.** (A-E) are genomics features and Fig 3. (F-J) are image features.

**Table 6. Features with significant p-values characterizing the low-high risk samples stratification in Kaplan Meier curve.**

| The top five most significant distinguishing genomics features | | | The top five most significant distinguishing image features | | |
|---|---|---|---|---|---|
| Features | Fig Caption | p-values | Features | Fig Caption | p-values |
| v2611 | A | 1.13E-257 | v540 | F | 1.81E-114 |
| v2324 | B | 7.49E-256 | v869 | G | 2.70E-101 |
| v2356 | C | 1.28E-254 | v93 | H | 7.01E-85 |
| v2337 | D | 1.14E-248 | v0 | I | 1.99E-84 |
| v2489 | E | 1.14E-248 | v173 | J | 8.49E-80 |

## 4. Discussion

Cancer grouping has a vast practical application in early detection, targeted therapies and survival risk management. Research outcomes have shown that different molecular datasets are now available and have been studied for better comprehension and implementation of the mentioned cancer group application. Most available molecular datasets still required the development of novel computational techniques for the extraction of essential information leading to improved survival of colon cancer patients. Some of the major challenges requiring urgent computation techniques tools are highly related to understanding molecular states of the available molecular data such as the genetic mutation, microsatellite instability status, co-expression pattern, biological markers, methylation, and several other molecular states synonymous with different cancer stages or groups. Novel methods are needed for the analysis of several molecular states identified in the unimodal and multimodal datasets for effective and robust cancer stages early detection, targeted therapies, and survival risk management. Some of the benefits of cancer stage accurate prediction include the determination of the best treatment for the patient, estimating the chances of the cancer returning or spreading after initial treatment, determining the chances of patient recovery and the best clinical trial options. Also, cancer staging helps in verifying how new treatments work among large groups of patients with the same diagnosis.

Colon cancer stages classification and prognostic require information about the tumor size, extent of spread of the disease to the nearest lymph nodes and metastasis to a distance site which are necessary for clinical treatment and survival. Our fused features from histopathological whole slide images and functional genomics provide a complementary model for effective stages prediction. The resulting fused features' latent space vectors aggregate microenvironment of tumors and molecular signals in both datasets. The result obtained from this research confirms previous result that genomics data represent the state-of-the-art for cancer stages prediction and the subjective nature of using histopathology images for cancer stage prediction. The 0.08% improved accuracy in cancer stage prediction in this research study indicates weak evidence contrary to expectation that integrated features will bring about significantly improved accuracy in cancer stages prediction.

Our result is consistent with findings obtained from previous studies showing that genomics features outperform images features predictive accuracy. Whereas the multimodal features from our studies did not give a clear improvement in predictive accuracy, others have shown similar trend or mediocre improvement in prediction accuracy. Examples of previous studies related to ours are in prediction of molecular subtypes of human breast cancer using multimodal data integrated from histopathological image, CNV and gene expression data. Also, studies on cancer prognosis prediction with multi-modal (histopathological images and mRNA) and multimodal representation for pan-cancer prognosis prediction [34–36] respectively.

Although our proposed deep neural network predictive model did not give expected differences in prediction accuracy between multimodal and genomics features in cancer stages prediction, 68% of the extracted features significantly had stratified the samples into low or high-risk survival group. These are features that are critical in informing viable cancer stages prediction and for analyzing risk factors within each survival group. While several constraints are encountered during the study, there are ways to improve them. First, only three genomics datasets are considered and therefore, extending the study to include other genomics and transcriptional dataset such as somatic mutation (SM), Copy number variation (CNV), and Reverse Phase Protein Array (RPPA) data expression could result in higher prediction accuracy and risk stratification. Also, the algorithm for integrating both features may have an effect on the performance of the classifier. Hence consideration and implementation of other integration algorithms are areas to be considered in future studies, particularly improvement in algorithm on selection of images tiles based on percentage of tumor content when image features only are used for predictive purpose. Another notable limitation is the absence of another independent datasets that involve both image and genomics data from the same samples, to verify and test our proposed framework. This will also be considered in future study. Stratification of samples into low and high-risk survival was based on the median values of each extracted feature which can be considered a deterministic approach, this could be compared with a stochastic approach of stratification in a future study.

Other cancer research areas that might benefit from our proposed framework include the classification of cancer of unknown, discernment of tumor microenvironment and microsatellite instability. In conclusion, extracted features that significantly stratified samples into clearly delineated low-high risk survival groups in this research study could be examined or linked with new or existing cancer biomarkers useful in diagnosis, prognosis and therapeutics cancer treatment in future research.

## Author Contributions

**Conceptualization:** Zeyneb Kurt.

**Data curation:** Olalekan Ogundipe.

**Formal analysis:** Olalekan Ogundipe.

**Investigation:** Olalekan Ogundipe.

**Methodology:** Olalekan Ogundipe.

**Supervision:** Zeyneb Kurt, Wai Lok Woo.

**Visualization:** Olalekan Ogundipe.

**Writing – original draft:** Olalekan Ogundipe.

**Writing – review & editing:** Olalekan Ogundipe, Zeyneb Kurt, Wai Lok Woo.

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
