## [Decision Letter · Decision Letter 0]

6 Feb 2024

PONE-D-23-30061Ensemble Deep Neural Networks (EDCNN) integrating genomics and histopathological images for predicting stages and survival time-to-event in colon cancer.PLOS ONE

Dear Dr. Kurt,

Thank you for submitting your manuscript to PLOS ONE. After careful consideration, we feel that it has merit but does not fully meet PLOS ONE’s publication criteria as it currently stands. Therefore, we invite you to submit a revised version of the manuscript that addresses the points raised during the review process.

We look forward to receiving your revised manuscript.

Kind regards,

John Adeoye

Academic Editor

PLOS ONE

Journal Requirements:

Additional Editor Comments:

Authors should consider revising their manuscript according to the comments provided by reviewers.

Reviewers' comments:

Reviewer's Responses to Questions

**Comments to the Author**

1. Is the manuscript technically sound, and do the data support the conclusions?

Reviewer #1: Yes

Reviewer #2: Yes

Reviewer #3: Yes

2. Has the statistical analysis been performed appropriately and rigorously? 

Reviewer #1: Yes

Reviewer #2: Yes

Reviewer #3: Yes

3. Have the authors made all data underlying the findings in their manuscript fully available?

Reviewer #1: No

Reviewer #2: Yes

Reviewer #3: No

4. Is the manuscript presented in an intelligible fashion and written in standard English?

Reviewer #1: Yes

Reviewer #2: Yes

Reviewer #3: No

5. Review Comments to the Author

Reviewer #1: Thank you very much for sharing this interesting work that could add to the body of existing works on the study of multimodal data for prediction of patient diagnosis, and outcomes. The authors applied AI methods to investigate the use of three types of genomic data, and histopathology images for predicting cancer stage, and survival risk in colon cancer patients.

Introduction:

“According to World Health Organization 2020 report, cancer is the leading cause of death accounting for over ten million deaths”. Please provide a citation for this statement.

“The complex features comprising the tumor size, the extent at which the cancer has spread to nearby lymph node, and whether the cancer has metastasized, which is the extent the cancer has spread to other parts of the body from the primary tumor still require the development of an efficient and effective model for accurate and improved classification and effective treatment management.” Please provide more clarity to this statement to make it obvious to a wider audience that the features mentioned are components of TNM staging. Since staging is the focus of this work.

Genomic Datasets preprocessing:

“Then an imputation function in python is used to fill out the missing values.” Please clarify if the missing values are filled with zero, or otherwise.

H&E image dataset preprocessing

"we build a sub python function called MX to filter out patches with less than 30% cellular tumor content." Please, how does this function work to select the patches with >30% cellular content.

Ensemble Deep Convolution Neural Network (EDCNN) Implementation:

The steps described in this section seem more like features integration than ensemble of models output. Also, this applies to the title of the manuscript.

Classification model:

“We test the validity of our hypothesis after reducing the dimension and consequently learn a higher-dimensional and compressed representation from the genomics and histopathological images.” Please provide more clarity about the higher-dimensional representation referred to here.

The described CNN architecture in this section seem more like an ANN.

Survival risk stratification with extracted features:

“As shown in Fig 3, the wide gap between the two functions is an indicator that we can confidently argue that 68% of the extracted features conveniently group the samples into low or high survival class.” Fig 3 only shows the result of 10 of 1836 features.

Is it possible to find out how many genomic, or image features are part of the 1836 for stratifying patients into risk groups?

Table 5:

Please explain more on the confusion matrix analysis. The number of samples in the table are not the same as those in tables 1, and 2.

Reviewer #2: This article has established an integrated deep neural network for the staging classification of colon cancer,stratifying samples into low or high-risk survival groups.There is unexplained diverse variation within the predefined stages of colon cancer. This variation can be observed when utilizing only features from either genomics or histopathological whole slide images as prognostic factors. The unraveling of this variation is expected to lead to enhanced staging and improved treatment outcomes.

Here are my suggestions:

1.This article has a certain degree of novelty.The novelty of colon cancer group can be made by the data of mRNA, miRNA and DNA methylation make,but if the practical application of this group is further elaborated, this paper will be more perfect.

2.If there is more clear introduction on how the randomness of the data is assured,the article will be better.

3.The model established in this article was applied data from open databases without external data validation. It is desirable that more data are used to verify the accuracy of the model in further research,the application scope of this model will be wider.

Reviewer #3: The authors have presented an important study integrating genomics and histopathological images for predicting stages and survival time-to-event in colon cancer. The manuscript is well written. I have the following comments:

1. The methodology section needs to be put in small headings to allow a proper understanding of the methodology. This will also allow for the reproducibility of your approach.

2. The authors mentioned that they have used 5 different sets of data from the Cancer Genome Atlas. These sets should be mentioned. Was the data from the same source? I mean if the data is from XXX Centre, does it contain all the datatypes (Clinical, DNA methylation, miRNA, mRNA, Hematoxylin, and Eosin (H&E) stained histopathological images). Hence, the specific source from TCGA should be mentioned since it is a database with numerous data.

3. The ethical permission statement should be mentioned.

4. The genomics resulted in 448 samples of clinical records with 80 features. What was the source of this data?

5. Also, 328 samples of mRNA expression with 20,502 features, 261 samples of miRNA expression with 1,870 features, and 353 samples of DNA methylation with 20,759 features. What was the specific source?

6. Regarding this statement, “We proceeded to get the Hematoxylin and Eosin (H&E) stained histopathological images for the 255 samples that have data in the genomics but got only 177 equivalent samples as in the genomics with images datasets”. Does it mean that all 177 samples were considered for clinical, DNA methylation, miRNA, mRNA, Hematoxylin, and Eosin (H&E) stained histopathological images?

7. Does it mean that the deep learning model development was based on data from Table 2?

8. This is unclear: (i) First, a biological feature (in any of methylation or mRNA or miRNA data) is removed, if more than 20% of the patients have a 0 value for it.

9. “Then an imputation function in python is used to fill out the missing values.” What imputation function was used? How many of the 177 were based on imputed data? How effective is the imputation approach. Perhaps, it may be better to remove the rows that contain missing values.

10. We input 36,926 merged features from miRNA, mRNA and DNA Methylation into autoencoder (AE) neural network design specifically to encode its input. How does the author arrive at 36, 926 features?

11. Were the 652 extracted features combined with 2,048 salient features from deep learning extraction?

12. The manuscript can also benefit from professional English Language proofreading.

13. The programming code for these analyses should be inserted as an Appendix/Supplementary for reproducibility.

6. PLOS authors have the option to publish the peer review history of their article (what does this mean?). If published, this will include your full peer review and any attached files.

Reviewer #1: No

Reviewer #2: No

Reviewer #3: No

---

## [Author Response · Author response to Decision Letter 0]

26 Mar 2024

Detailed and one-to-one responses to the reviewer comments have been provided in the document uploaded and entitled 'Response to Reviewer comments PONE-D-23-30061.docx'

---

## [Decision Letter · Decision Letter 1]

3 May 2024

PONE-D-23-30061R1Deep Neural Network (DNN) integrating genomics and histopathological images for predicting stages and survival time-to-event in colon cancerPLOS ONE

Dear Dr. Kurt,

Thank you for submitting your manuscript to PLOS ONE. After careful consideration, we feel that it has merit but does not fully meet PLOS ONE’s publication criteria as it currently stands. Therefore, we invite you to submit a revised version of the manuscript that addresses the points raised during the review process.

We look forward to receiving your revised manuscript.

Kind regards,

John Adeoye

Academic Editor

PLOS ONE

Journal Requirements:

Reviewers' comments:

Reviewer's Responses to Questions

**Comments to the Author**

1. If the authors have adequately addressed your comments raised in a previous round of review and you feel that this manuscript is now acceptable for publication, you may indicate that here to bypass the “Comments to the Author” section, enter your conflict of interest statement in the “Confidential to Editor” section, and submit your "Accept" recommendation.

Reviewer #1: (No Response)

Reviewer #3: All comments have been addressed

2. Is the manuscript technically sound, and do the data support the conclusions?

Reviewer #1: Yes

Reviewer #3: Yes

3. Has the statistical analysis been performed appropriately and rigorously? 

Reviewer #1: Yes

Reviewer #3: N/A

4. Have the authors made all data underlying the findings in their manuscript fully available?

Reviewer #1: Yes

Reviewer #3: Yes

5. Is the manuscript presented in an intelligible fashion and written in standard English?

Reviewer #1: Yes

Reviewer #3: Yes

6. Review Comments to the Author

Reviewer #1: Thank you very much for your response to the previous comments, and for the changes made.

As regards the the confusion matrix on table 5, the table shows a total of 112,161 samples that were predicted by the models. This applies to each of genomic, images, and the integrated data. However, only image samples contain a total of 112,161 tiles. On the patient level, the genomic data contain 177 samples, and same for image data too. If this is correct, I would expect at least the the number of genomic, and integrated samples to be 177. Please, can you clarify more on these? Were the image tiles eventually aggregated to patient level? what methods was used for the aggregation?

Reviewer #3: The authors have addressed my comments. I have no further comments. I wish the author the best in their research.

7. PLOS authors have the option to publish the peer review history of their article (what does this mean?). If published, this will include your full peer review and any attached files.

Reviewer #1: No

Reviewer #3: No

---

## [Author Response · Author response to Decision Letter 1]

23 May 2024

Our point-to-point responses to the reviewers’ comments are provided in the document entitled “Response to Reviewer comments PONE-D-23-30061R1.docx”.

---

## [Decision Letter · Decision Letter 2]

28 May 2024

Deep Neural Networks integrating genomics and histopathological images for predicting stages and survival time-to-event in colon cancer

PONE-D-23-30061R2

Dear Dr. Kurt,

We’re pleased to inform you that your manuscript has been judged scientifically suitable for publication and will be formally accepted for publication once it meets all outstanding technical requirements.

Kind regards,

John Adeoye

Academic Editor

PLOS ONE

Additional Editor Comments (optional):

Reviewers' comments:

Reviewer's Responses to Questions

**Comments to the Author**

1. If the authors have adequately addressed your comments raised in a previous round of review and you feel that this manuscript is now acceptable for publication, you may indicate that here to bypass the “Comments to the Author” section, enter your conflict of interest statement in the “Confidential to Editor” section, and submit your "Accept" recommendation.

Reviewer #1: (No Response)

2. Is the manuscript technically sound, and do the data support the conclusions?

Reviewer #1: Yes

3. Has the statistical analysis been performed appropriately and rigorously? 

Reviewer #1: Yes

4. Have the authors made all data underlying the findings in their manuscript fully available?

Reviewer #1: Yes

5. Is the manuscript presented in an intelligible fashion and written in standard English?

Reviewer #1: Yes

6. Review Comments to the Author

Reviewer #1: Thank you very much for the clarification about many to one method of concatenation used. About table 5, to make it clearer, it may be better if the number of genomic sample is presented as 35, and not 22432 as it is with image, and integrated data. Currently, it seems like the same genomic samples were repeated many times in both the training and test data.

7. PLOS authors have the option to publish the peer review history of their article (what does this mean?). If published, this will include your full peer review and any attached files.

Reviewer #1: No

---

## [Editor Report · Acceptance letter]

22 Jul 2024

PONE-D-23-30061R2 

PLOS ONE

Dear Dr. Kurt, 

I'm pleased to inform you that your manuscript has been deemed suitable for publication in PLOS ONE. Congratulations! Your manuscript is now being handed over to our production team.

Kind regards, 

on behalf of

Dr. John Adeoye 

Academic Editor

PLOS ONE